environmental science/environmental engineering

POI, Luojia1-01, multi-source data fusion, urbanization, urban construction

**Author for correspondence:**
Zijiang Yang
e-mail: kmzjy@126.com

# Extraction of urban built-up area based on the fusion of night-time light data and point of interest data

Xiong He[1,2,3], Zhiming Zhang[2] and Zijiang Yang[3]

[1]School of Geography and Planning, Sun Yat-sen University, Guangzhou 510275, People's Republic of China
[2]School of Ecology and Environmental Science, Yunnan University, Kunming 650031, People's Republic of China
[3]School of Architecture and Planning, Yunnan University, Kunming 650031, People's Republic of China

XH, 0000-0002-6848-8327; ZZ, 0000-0002-8037-0559;
ZY, 0000-0001-7180-6974

The accurate extraction of urban built-up areas is an important prerequisite for urban planning and construction. As a kind of data that can represent urban spatial form, night-time light data has been widely used in the extraction of urban built-up areas. As one of the geographic open-source big data, point of interest (POI) data has a high spatial coupling with night-time light data, so researchers are beginning to explore the fusion of the two data in order to achieve more accurate extraction of urban built-up areas. However, the current research methods and theoretical applications of the fusion of POI data and night-time light data are still insufficient compared with the dramatically changing urban built-up areas, which needed to be further supplemented and deepened. This study proposes a new method to fuse POI data and night-time light data. The results before and after data fusion are compared, and the accuracy of urban built-up area extracted by different data and methods is analysed. The results show that the data fusion can avoid the shortage of single data and effectively improve the extraction accuracy of urban built-up areas, which is greatly helpful to supplement the study of data fusion in urban built-up areas, and also can provide decision-making guidance for urban planning and construction.

## 1. Introduction

In recent decades, with the rapid urbanization of China and the intensification of agricultural modernization in urban built-up areas, cities have undergone unlimited expansion and great changes have also occurred [1,2]. Urban built-up areas are the

carrier of all urban activities, the main gathering areas of population and economic activities, and the prominent manifestation of urbanization [3]. The implementation of 'national spatial planning' in China clearly proposes the demarcation of the boundary line of urban areas [4,5]. Therefore, the extraction of urban built-up areas has become increasingly important.

At present, the extraction and division of urban built-up areas in China mainly depend on the guidance of the government's statistical yearbook and planning documents, but these cannot truly and objectively reflect the scope of urban built-up areas [6,7]. This study hopes to further improve the extraction effect of urban built-up areas through data fusion on the basis of existing research.

The urban built-up area refers to the non-agricultural construction land with complete functions of municipal public facilities within the urban area. The urban built-up area is the main carrier of urban population activities in urban space and a prominent manifestation of the high concentration of population and economy [8,9]. To some extent, the number of urban built-up areas determines the level of regional development. Therefore, the accurate extraction of urban built-up areas is not only helpful to understand the current situation of urbanization [10,11], but also has a great correlation with the development of cities [12]. With the acceleration of global urbanization, urban built-up areas expand rapidly in a short period of time, accompanied by urban problems such as land-use change and population imbalance caused by urban built-up area expansion [13,14]. Therefore, accurate extraction of urban built-up areas is also of great importance to alleviate urban problems in the process of urbanization [15,16].

Night-time light remote sensing data, which reflects the function of urban infrastructure through the brightness of urban night-time lights, is the remote sensing data widely used in relevant urban research [17,18]. To some extent, night-time light can make up for the lack of panel statistical data in urban space research. What's more, night-time light can be applied in the study of urban space because urban activities are closely correlated with electricity consumption [19,20], and many studies have proved that the intensity of urban light has a high correlation with urban population distribution [21]. Therefore, night-time light data is currently mainly used in urban expansion [22], urban morphology and structure [23], and social and economic status judgement [24].

In the research of urban spatial structure, researchers tested the relationship between urban residential areas and rural areas by using the data of the Defence Meteorological Program Operational Line-Scan System (DMSP/OLS) [25]. Yu *et al.* [26] developed a new object-oriented method in 2014 to characterize the urban model for the analysis of night satellite images; moreover, VIIRS (Suomi National Polar-orbiting Partnership/Visible Infrared Imaging Radiometer Suite) data makes it possible to examine the urban internal spatial structure in more detail [27]. What's more, subsequent scholars have also conducted some studies on urban spatial structure with finer resolution on the basis of VIIRS data [28], including the re-evaluation of social and economic indicators and the detection of urban internal regional structure [29,30].

Luojia1-01 night-time light image, newly released by Wuhan University in 2018, has a spatial resolution of 130 m, which can more clearly reflect the night structure of the city [31]. Although the night-time light has relatively higher spatial stability and retains the integrity of the study area as much as possible, the sensor cannot record the socio-economic attributes and human daily activities [19,32,33]. What's worse, the sensor will not only record the strong light emitted by the city, but also the strong light emitted by the airport, road, port, etc., which further leads to the inaccurate judgement of the urban area [34,35].

With the further development of technology, more and more big data begins to be applied to the study of the city [36], including network review data [37], social software data [38], Internet data [39], thermal graph data [40], POI data [41], dating data [42], the Integrated Circuit Chip (IC) card data [43], mobile phone signal data [44], GPS data [45] and so on. The influences of these factors on the spatial connection [46], the functional layout [47] and the behavioural space of the city have been studied in detail [47]. These studies fully show that big data has a higher adaptability in urban space research than traditional data [48,49].

POI data is the expression of all urban entity abstraction in virtual geographic space, which has the advantages of high accuracy, fast data update speed and large data volume, etc [50], so it has been widely used as a new form of geospatial data in recent years [51,52]. At present, POI data are mainly used in urban spatial structure. At present, POI data are mainly used in the spatial structure of cities, including the identification of urban centres [53], the distribution of urban facilities [54], the division of urban functions [55] and the analysis of spatial patterns [56,57]. Compared with traditional data, although POI data research in urban space has advantages of higher speed and better accuracy, there are relatively few studies on the application of POI data in the identification of urban built-up areas [58]. Some scholars have identified the abrupt character of POI data in the city boundary [55] and

summarized the general threshold value of POI data to extract the city boundary in Chinese cities. POI data are also used to identify the main built-up areas of cities with high agglomeration in urbanization [59,60]. However, due to the high-density difference between the number of POI in built-up and non-built-up urban areas, it is easy to cause errors [57].

Data fusion has been widely used in the research of cities and related disciplines because it can not only integrate the advantages of multiple sources of data, but also better reflect the outstanding advantages of single data [58]. It has been shown in the studies that night-time light data and POI data have a strong coupling in urban space [61], so the fusion of the two data has a good effect in the application of urban space, including urban space identification, urban built-up area extraction, etc. By comparing the research of single-source data [60], it can be found that although the fusion of night-time light data and POI data can significantly improve the accuracy of urban spatial identification and urban built-up area extraction, the research on data fusion is still insufficient compared with the drastic changes of urban built-up areas [62,63]. Therefore, in order to extract the urban built-up area more accurately and supplement the related research of data fusion, this study attempts to extract the urban built-up area of Kunming, China by fusing night-time light data and POI data and uses Guangzhou, China as a case for verification and analysis to explore the extraction accuracy of built-up areas in this study. Finally, a reliable method and path for the accurate extraction of urban built-up areas by the fusion of night light and POI data are obtained. Compared with other studies, this study first analyses the case of Kunming, and then uses Guangzhou as a case for further supplementary analysis to explore the research and application of data fusion in urban built-up area extraction, which complements the theoretical and practical application of data fusion in urban built-up area extraction, and plays a good role in promoting the research of urban built-up areas and urban space. It also has a practical and positive guiding role for future urban planning and construction.

# 2. Material and methods

## 2.1. Study area

Kunming, the provincial capital of Yunnan province, is one of the most urbanized cities in western China. With the development and promotion of national macro strategies such as 'One Belt and One Road' [64] and 'western development', the urban built-up area of Kunming has expanded significantly in recent years [65]. According to the statistical yearbook of Yunnan province in 2019, Kunming covers an area of $21\,473\,km^2$, with an urban built-up area of $437\,km^2$. Although the main urban area of Wuhua District, Guandu District, Xishan District and Chenggong District is only $2542\,km^2$, the urban built-up area reaches $434.4\,km^2$, which is also the reason why the five main urban areas of Kunming City are selected as the study area (figure 1) [50,66].

## 2.2. Research data

### 2.2.1. Luojia1-01

The night-time light data used in this study is derived from the Luojia1-01 scientific experiment satellite launched by Wuhan University of China, with a spatial resolution of 130 m. The main reasons for choosing Luojia1-01 night-time light data in this study are as follows: first, Luojia1-01 night-time light data greatly improves the time and spatial resolution of night-time light data. Second, it improves some problems of traditional night-time light including low resolution of DMSP/OLS and NPP/VIIRS data and light overflow, etc. In addition, it also provides light data for the identification of individual cities and small and medium-sized urban built-up areas. As for the source of the LuoJia-1A data used in this study, it is mainly obtained from the Hubei Data and Application Network of the high-resolution earth observation system. The main data used in this study is the Luojia1-01 night-time light data for March 2019 published on this website and shown in figure 1.

### 2.2.2. Point of interest data

POI refers to all urban entities abstracted in geographic space. Each point of interest (POI) contains the attribute categories and longitudes and latitudes of the abstract urban entities, which basically cover all aspects of urban functions due to its huge number. The POI data used in this study is mainly from Amap, which, as one of the three major map service providers in China, provides the most extensive and

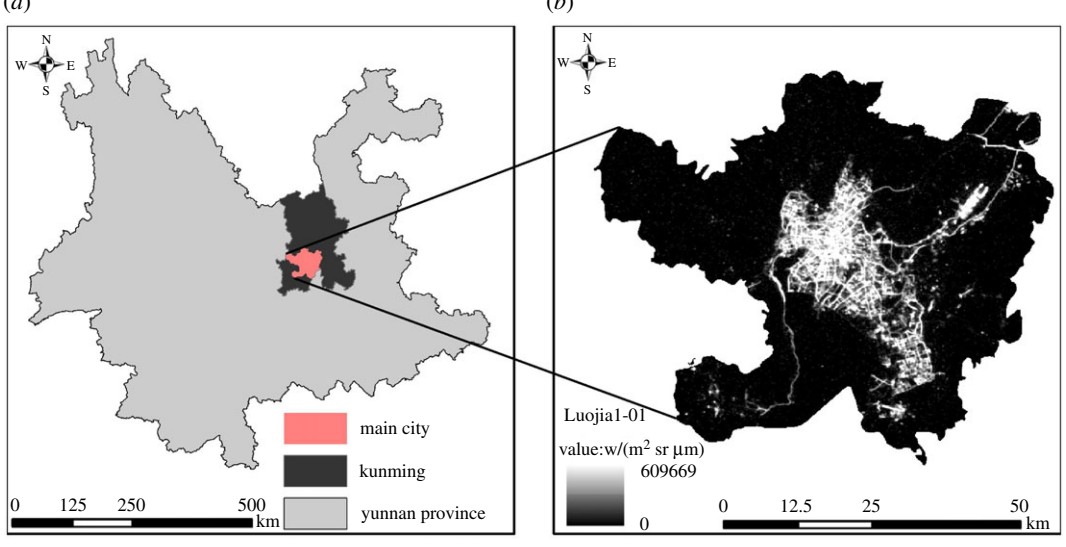

**Figure 1.** Study area.

**Table 1.** POI classification of Kunming.

| primary classification | primary classification | Number/% |
|---|---|---|
| public service | roads, public facilities, traffic facilities, access facilities, government agencies, science, education and culture | 224 019/16.74 |
| life service | catering services, shopping services, healthcare | 826 657/6175 |
| recreation and entertainment | recuperation and vacation, entertainment related, sports venues | 49 442/3.69 |
| living | community, villa, dormitory | 45 621/3.41 |
| commercial affairs | companies, enterprises, banks | 192 871/14.41 |
| sum | | 1 338 610/100 |

relatively accurate positioning services. In this study, with the help of the API interface provided by Amap, a total of 133 861 0 941 POI data are obtained from the main urban area of Kunming in March 2019 as shown in table 1.

### 2.2.3. Verification data

A total of 3000 pixel points are randomly generated in Kunming and Guangzhou, which are further determined to be located in the study area by data screening. In addition, the Google Earth high-definition map is used to further identify the random points in Kunming and Guangzhou. It is found that the number of random points in Kunming's non-built-up area is 2231, and the number of random points in the built-up areas is 769, while the number of random points in Guangzhou's non-built-up area is 1466, and the number of random points in built-up areas is 1534.

## 2.3. Methods

### 2.3.1. OSTU threshold segmentation method based on grey level and gradient mapping function

In the previous studies on the extraction of night-time light from urban built-up areas, different methods are often used to determine the extraction threshold of night-time light. The reason for this is that the accurate selection of night-time light threshold is considered to be the most important. According to the binarization characteristic (grey characteristic) of the image itself, the OSTU algorithm can divide the image into two parts: segmentation scene (foreground) and reference scene (background). Moreover,

this algorithm can further judge the segmentation threshold of the automatically selected image by binarization of the whole image. When the threshold of image segmentation is selected, the difference between foreground and background is the biggest, which is also the segmentation standard of the OSTU algorithm [67]. While this study attempts to determine the threshold value from the pixel value of night-time light image, and the OSTU threshold segmentation method is considered to be a very good algorithm for image threshold segmentation, the reasons are as follows: first of all, the OSTU threshold can be calculated by MBV (maximum between-cluster variance) image; second, the OSTU algorithm has the advantages of simple calculation, fast operation and effective results, etc. Therefore, this study hopes to use the OSGM algorithm based on grey level and gradient mapping (GGM) function to detect the extraction threshold of night-time light [68,69]. If $\alpha$ stands for the greyscale value of the image and $\beta$ for the average gradient value of the image, then the mapping function $\beta = s(\alpha)$, where the mapping relation (·) stands for the average gradient value level $\alpha$ of all pixels with greyscale.

Suppose that the function $f(x, y)$ stands for the greyscale value of the pixel whose coordinate is $(x, y)$ in the image, and its gradient value is defined by the two-dimensional vector as follows:

$$\nabla f = \left[\frac{\partial f}{\partial x} \frac{\partial f}{\partial y}\right], \tag{2.1}$$

the calculated value of the two-dimensional vector is

$$\nabla f = \mathrm{mag}(\nabla f) = [G_x^2 + G_y^2]^{1/2} = \left[\left(\frac{\partial f}{\partial x}\right)^2 + \left(\frac{\partial f}{\partial y}\right)^2\right]^{1/2}. \tag{2.2}$$

Suppose that the function $G(x, y)$ stands for the gradient function of the image, then $G(x, y) = \nabla f$. Suppose that the number of pixels with greyscale i (i∈ [0; 1, … L −1], where L is the existing greyscale level) is $n_i$, then its pixel set should be expressed as

$$R_i = \{(x, y)|f(x, y) = i\}. \tag{2.3}$$

Then, grey level and gradient mapping are defined as follows:

$$T(i) = \frac{\sum_{(x,y)\in R_i} G(x, y)}{n_i} i \in [0,1,\cdots,L - 1]. \tag{2.4}$$

The calculation step of the OSTU algorithm is to first calculate the gradient value $f(x, y)$ of each pixel of the image and then calculate the average gradient value $T$ of the pixels with the same grey level in the image. Finally, the grey value $i$ of the maximum difference between the foreground and the background is calculated after setting the function of the GGM function. The obtained greyscale $i$ is the optimal image segmentation threshold.

### 2.3.2. Density-graph

The spatial structure inside the city can be directly reflected by the density distribution of POI in the urban space. Since the number of economic activities and public service facilities in the non-built-up area is far less than that of the urban built-up area, the density distribution of POI will decrease from the built-up area to the non-built-up area, and the density of POI will show an irreversible downward trend at the junction between the built-up area and the non-built-up area [60].

The critical value of POI number mutation is determined by constructing the density-graph of the POI point density curve value $d$ and the theoretical area $S_d$ of the closed contour, the theoretical radius $\Delta s_d \wedge (1/2)$. The solution steps of density-graph are mainly divided into two steps, the first step is to determine the relationship between the density value $d$ and the theoretical radius increment $\Delta s_d \wedge (1/2)$ and to obtain the derivative of the theoretical radius increment $\Delta s_d \wedge (1/2)$. For the theoretical radius increment $\Delta s_d \wedge (1/2)$ is derived, the equation $d$ holds in theory. If the equation is equal to 0, then the density curve is uniformly diffused outwards. However, it is well known that there is basically no uniform expansion mode in urban built-up area expansion. Therefore, if the equation is greater than 0, the density curve is diffused outwards; if the equation is less than 0, the density curve is contracted inwards. The second step is to judge the critical value of density-graph calculation. Actually, urban is a very complex system. As most cities experience non-uniform outward expansion from multiple city centres or clusters of cities, the Kernel Density Curve of point elements changes dramatically inside the urban space, which is also the reason why the performance of the

density-graph curve inside the city should be fluctuating. However, from the macro scale of the whole urban space, the fluctuation of the density-graph curve should have a critical point with global significance. When $\lim((d(\Delta S_d \wedge (1/2))/dd)) > r$ appears, that is, when the fluctuation of density-graph presents an irreversible growth trend, it can be considered that $r$ is a critical point with global significance, which can also be considered as the critical value of the urban built-up area.

if

$$\lim \frac{d(\Delta S_d \wedge (1/2))}{dd} > r. \tag{2.5}$$

Then, $r$ is the critical point of urban built-up area and non-built-up area identified by POI.

### 2.3.3. Data fusion

There are shortcomings in only using night-time light and only using POI to identify urban built-up areas. For example, there are problems such as low spatial resolution and light overflow when only using night-time light to identify urban built-up areas, while only using POI would form too high-density difference between built-up and non-built-up areas. This is the reason why this study attempts to integrate the two kinds of data into a new comprehensive data to identify urban built-up areas [61].

In this study, 'geometric mean' is used to integrate night-time light data and POI data. The reason for this is that there is a considerable amount of difference between the POI kernel density value and Digital Number (DN) value of night-time light and there are noise points in the night-time light data. While the use of geometric mean in image fusion can effectively eliminate the impact of image extremum and retain the original information of image, so it is widely used in image fusion [70,71]. The LJ&POI data after fusion can eliminate the difference of order of magnitude caused by the too large density analysis value. Meanwhile, to some extent, it can eliminate the background noise of night-time light image and reduce the influence of light overflow. The calculation formula is as follows:

$$POIDN_i = \sqrt{POI_i \times DN_i}, \tag{2.6}$$

where $POIDN_i$ stands for the composite index, $POI_i$ for the POI kernel density value of point $i$ and $DN_i$ for the night-time light brightness value of point $i$.

The fraction is the harmonic average of recall and precision with a value range of 0–1. The higher the value is, the higher the precision is. The recall rate, precision rate and $F_1$ score are, respectively, as follows:

$$\text{precision} = \frac{a_{\text{overlap}}}{a_{\text{computed}}}, \tag{2.7}$$

$$\text{recall} = \frac{a_{\text{overlap}}}{a_{\text{comparative}}} \tag{2.8}$$

$$\text{and} \quad F_1 = \left(\frac{2}{\text{recall}^{-1} + \text{precision}^{-1}}\right) = 2\,\frac{\text{precision recall}}{\text{precision} + \text{recall}}, \tag{2.9}$$

$a_{\text{overlap}}$ stands for the total area of the overlapped part of the built-up area and the reference built-up area, $a_{\text{computed}}$ for the total area of built-up area extracted, $a_{\text{comparative}}$ is the total area of the reference built-up area.

The flow chart of this study is shown in figure 2.

## 3. Results

### 3.1. Urban built-up areas identified by different data

#### 3.1.1. Urban built-up areas identified by point of interest data

The spatial distribution of POI in Kunming is obtained by point density analysis of POI. It can be concluded that POI in the main urban area of Kunming presents a spatial distribution pattern of decreasing circles. The high POI density is mainly distributed in the regions with higher urbanization level. With the decline of urbanization level, the POI density also shows a decreasing trend. From the perspective of the overall spatial distribution of POI point density in the study area, the distribution of POI in urban space has an obvious rule, which is that the POI density changes from dense to sparse from the urban centre to the urban edge.

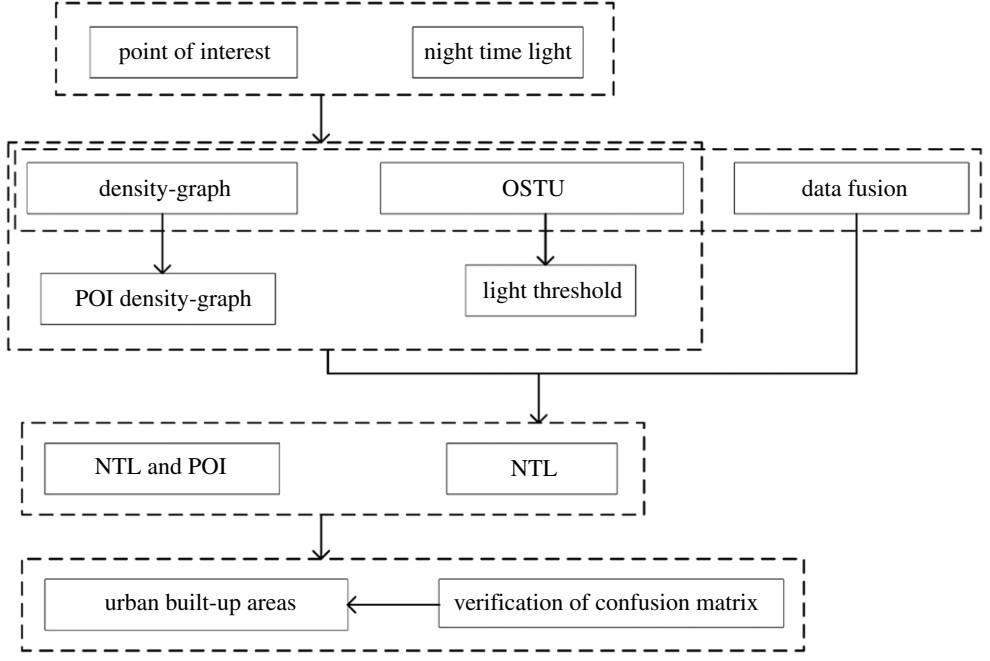

**Figure 2.** Analytical framework.

The ratio relationship between POI density and theoretical radius is calculated by constructing density-graph. It is concluded that when the theoretical radius of POI density is 772, the POI density showed an irreversible downward trend. That is, when $r = 772$, the area enclosed by the POI density radius is the urban built-up area.

The urban built-up area identified by POI is 381.68 km$^2$ and the circumference is 415.5 km. The ratio of area to circumference is 1.08. The boundary of the extracted urban built-up area is of low complexity. Although the number of patches extracted from urban built-up areas is as many as 41, the internal details of urban built-up areas are not reflected in the patches of large urban built-up areas, and the actual situation of urban built-up areas is not reflected in the patches of scattered urban built-up areas although the plaque separation degree here is high. As can be seen from figure 3, the distribution pattern of urban built-up areas in Kunming conforms to the pattern of 'urban built-up areas—urban fringe areas and non-built-up areas'. Objectively speaking, the urban built-up areas based on POI identification generally generate smooth curves near the identified boundary, with the shortcomings of simple edge details and serious information loss; therefore the results of identification do not conform to the actual urban built-up areas.

### 3.1.2. Urban built-up areas identified by Luojia1-01 data

It can be seen from the night-time light graph of Luojia1-01 in Kunming that Luojia1-01 can clearly sketch the outline of urban built-up area in Kunming, and the brightness of night-time light is obviously positively correlated with the urbanization level of the region. However, due to the high light value in the airport and the serious problem of light overflow in the whole city, it is difficult to determine the segmentation threshold of night-time light just by observing. It can be calculated by the OSTU algorithm based on GGM function that the optimal segmentation threshold of night-time light of Luojia1-01 in Kunming city is $T(i) = 0.00063$, and the urban built-up area extracted by $T(i) = 0.00063$ is shown in figure 4.

The urban built-up area identified by Luojia1-01 is 391.84 km$^2$ and the circumference is 706.98 km. The ratio of area to circumference is 0.62, which shows that the complexity of the boundary of the extracted urban built-up area has been improved. Although the total number of plaques extracted in the urban built-up areas is as many as 96, most of the built-up area plaques are severely fragmented, and the identification of urban built-up areas is severely disturbed, except for the plaques of the urban built-up areas of the three major cities of Chenggong new town, the airport and the central city. It can be found from figure 5 that the edge details of urban built-up areas extracted by night-time light are too complex, and there is too much interference information inside. In addition, there is the problem of light overflow, which all interfere with the extraction of urban built-up areas.

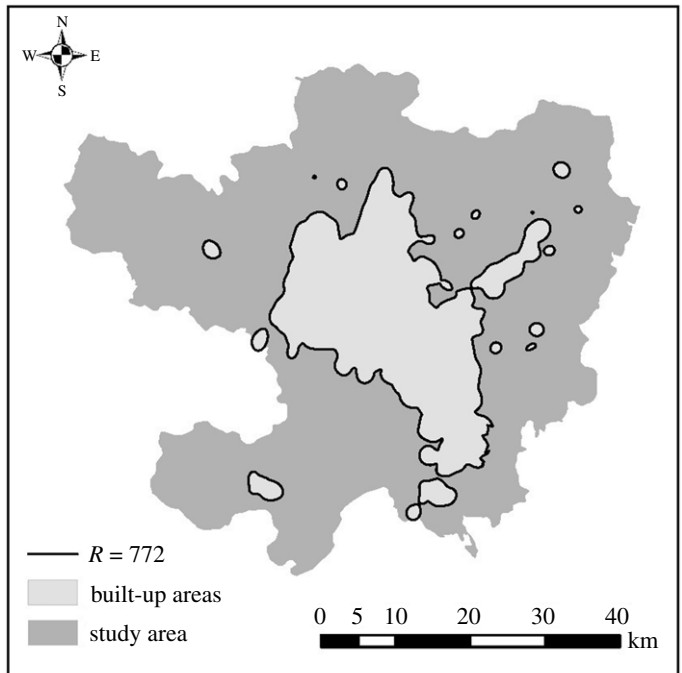

**Figure 3.** Urban built-up areas identified by POI data.

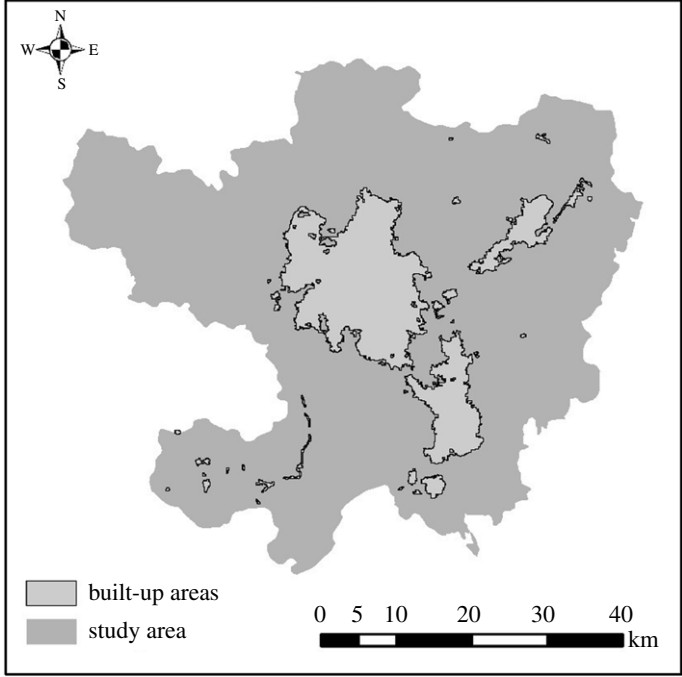

**Figure 4.** Urban built-up areas identified by Luojia1-01 data.

### 3.1.3. Urban built-up areas identified by data fusion

The new fusion data LJ&POI is obtained by the fusion calculation of the geometric mean value between the POI data and the Luojia1-01 data. It can be found through observation that LJ&POI data combines the advantages of two kinds of data: namely, it has the stability and effectiveness of POI data itself, while further eliminating the influence of light noise. Compared with Luojia1-01 data, LJ&POI data have significantly improved the problem of light spillover. In addition, the expression of internal details, including urban streets, is clearer and more complete. Compared with POI data, LJ&POI data have

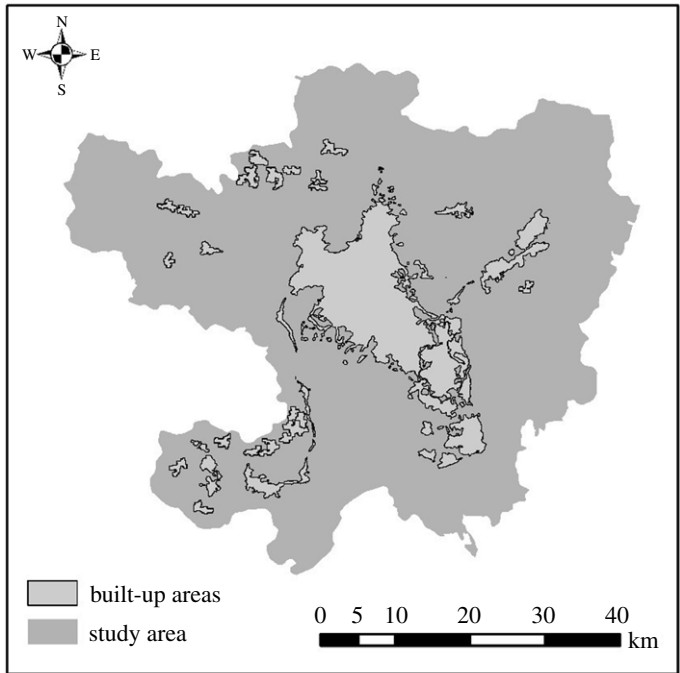

**Figure 5.** Built-up area identified by LJ&POI based on density-graph.

more real urban internal expression and richer urban boundary. In general, from the perspective of the data fusion effect, the fused LJ&POI data can more accurately express the urban built-up area.

By constructing the density-graph analysis of LJ&POI data and comparing the relationship between the added value of LJ&POI density and the theoretical radius, it is concluded that when $r = 738$, the density of LJ&POI presents an irreversible downward trend. The urban built-up area extracted based on $r = 738$ is shown in figure 5.

The area of the urban built-up area identified by LJ&POI based on the density-graph is 413.46 km² with a perimeter of 669.7 km. The ratio of area to circumference is 1.68, which shows that the complexity of the boundary of the extracted urban built-up area is a little high. Although the total number of urban built-up area plaques extracted is as many as 64, the urban built-up area plaques are well separated, and the internal details of the identified urban built-up area are clear, especially in the urban boundary zone. Due to the improvement of the complexity of the urban boundary, the effect of identification is closer to the real situation of the urban built-up areas and the urban built-up area is better expressed.

The segmentation threshold of LJ&POI can be calculated by using the OSTU algorithm based on GGM function. It can be seen from the algorithm calculation that when the segmentation threshold $T(i)$ is 672.38, the algorithm obtains the best result, so the area with a pixel value greater than 672.38 is the urban built-up area (figure 6).

The area of the extracted urban built-up area identified by the OSTU algorithm based on GGM function of LJ&POI is 405.05 km² with a perimeter of 635.65 km. The ratio of area to circumference is 1.51, which shows that the complexity of the boundary of the extracted urban built-up area is reduced. Although the total number of plaques extracted from the urban built-up area is as many as 52, the degree of plaque fragmentation in the urban built-up area is reduced, and the gaps in the urban built-up area are well compensated within the city. As can be seen from figure 6, the urban built-up area extracted based on LJ&POI can better express the urban built-up area, and the complex interference information of the boundary of the urban built-up area extracted from the fusion data is less, which all make the extracted built-up area closer to the real built-up area, and the internal details of the extracted built-up area closer to the real built-up area, and the inner details of the city are also improved.

## 3.2. Comparison of extraction results

Referring to the reference built-up area generated by the verification data, it can be found that the overlap area between the reference built-up area and the identified built-up area can be calculated on the

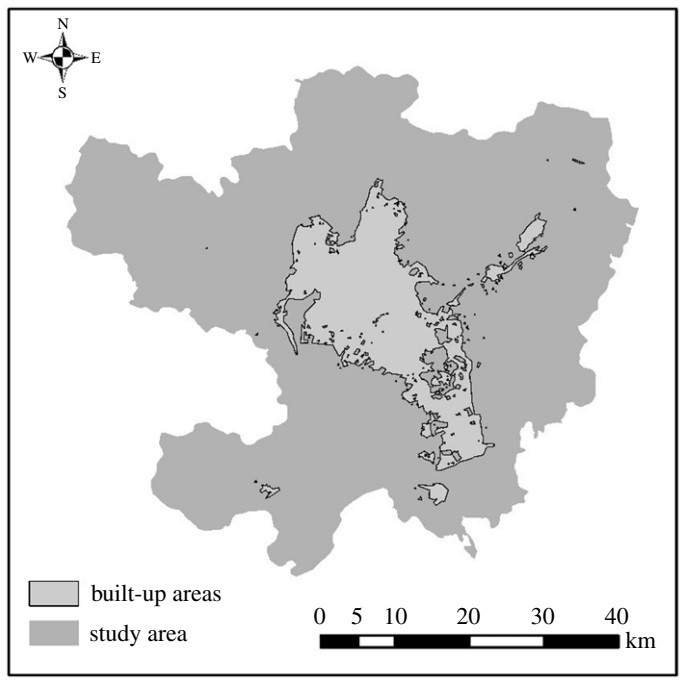

**Figure 6.** Built-up area identified by LJ&POI based on GGM function.

**Table 2.** Accuracy verification of urban built-up area results extracted from different data.

|           | POI    | Luojia1-01 | LJ&POI(D-G) | LJ&POI(OSTU) |
|-----------|--------|------------|-------------|--------------|
| recall    | 0.7466 | 0.5777     | 0.8405      | 0.8119       |
| precision | 0.6522 | 0.5215     | 0.8018      | 0.7589       |
| F1 score  | 0.6990 | 0.5478     | 0.8207      | 0.7845       |

geographical location. Through calculations, it can be found that the overlapping areas of POI, Luojia1-01, LJ&POI(D-G) and LJ&POI(OSTU) with the reference built-up area are 245.07 km$^2$, 226.6 km$^2$, 347.51 km$^2$ and 328.9 km$^2$, respectively. Among them, the built-up area extracted by LJ&POI(D-G) overlaps the reference built-up area the most, and the built-up area extracted by Luojia1-01 overlaps the reference built-up area the least.

The built-up areas extracted from different data accounted for 56.41%, 52.16%, 80.00% and 75.71% of the total reference area, respectively. With the accuracy verification of classification indexes, the recall rate, precision rate and F1 score of urban built-up areas extracted from POI data, Luojia1-01 data and LJ&POI data by different methods are calculated, respectively. The accuracy verification results are shown in table 2.

It can be concluded from table 2 that the accuracy results of urban built-up areas extracted from different data are LJ&POI(D-G) > LJ&POI(OSTU) > Luojia1-01(OSTU) > POI(density-graph), among which the verification results of LJ&POI(OSTU) and LJ&POI (density-graph) are similar, and the verification results of Luojia1-01(OSTU) and POI (density-graph) are similar.

After analysing table 2, it can be seen that the results of urban built-up areas extracted based on single data are not good either in the proportion of overlapping area or in the accuracy verification. POI data declines too fast in areas with low urbanization, which results in the actual urban built-up areas extracted at the city boundary and around Chenggong new town being far smaller than the actual urban built-up areas. At the same time, Luojia1-01 lighting data has obvious light overflow problems and light gaps within the city, which leads to a large difference between the extracted urban built-up area and the actual urban built-up area. Although the urban built-up areas extracted from the POI data and Luojia1-01 data all reach an area of more than 80%, according to the accuracy verification results, the extracted built-up areas are not highly coincident with the actual built-up areas geographically.

The results of urban built-up area and accuracy verification based on the fusion data LJ&POI, whether based on the density-graph or the OSTU algorithm of GGM, are better than the results of

**Table 3.** Confusion matrix and Kappa coefficient.

| data | | built-up area | non-built-up area | overall accuracy (%) | Kappa |
|---|---|---|---|---|---|
| Kunming POI | built-up area | 413 | 681 | 72.31 | 0.63 |
| | non-built-up area | 449 | 1583 | | |
| Guangzhou POI | built-up area | 434 | 638 | 72.20 | 0.64 |
| | non-built-up area | 405 | 1499 | | |
| Kunming Luojia1-01 | built-up area | 426 | 663 | 75.60 | 0.61 |
| | non-built-up area | 417 | 1494 | | |
| Guangzhou Luojia1-01 | built-up area | 432 | 624 | 71.90 | 0.66 |
| | non-built-up area | 411 | 1533 | | |
| Kunming LJ&POI(OSTU) | built-up area | 486 | 420 | 90.87 | 0.85 |
| | non-built-up area | 357 | 1737 | | |
| Guangzhou LJ&POI(OSTU) | built-up area | 513 | 459 | 90.72 | 0.83 |
| | non-built-up area | 330 | 1698 | | |
| Kunming LJ&POI(D_G) | built-up area | 753 | 234 | 89.80 | 0.89 |
| | non-built-up area | 90 | 1923 | | |
| Guangzhou LJ&POI(D_G) | built-up area | 801 | 111 | 91.12 | 0.87 |
| | non-built-up area | 42 | 2046 | | |

single data extraction. In terms of the built-up area extracted, it is increased to more than 90%, and the accuracy verification index is also increased by nearly 20%. Compared with the POI data, the LJ&POI data retains the attribute of night-time light data, which makes the extracted built-up area in Chenggong new town and the boundary of urban built-up area more detailed, and the extracted result is better; compared with the Luojia1-01 data, the urban built-up area extracted by LJ&POI data makes up for the urban light void well within the city and effectively improves the problem of light overflow, which all make the extraction results better.

### 3.2.1. Accuracy verification of extraction results

Accuracy verification is an important step to test the reliability of this method for urban built-up area extraction. In this study, the urban built-up areas of Guangzhou are extracted by the fusion of Guangzhou night-time light data and POI data; besides, the confusion matrix and overall accuracy are used to verify the reliability and practicability of data fusion for the extraction of urban built-up areas. The extracted urban built-up areas of Guangzhou and the confusion matrix and accuracy verification are shown in table 3.

It can be seen from table 3 that the highest accuracy value and Kappa coefficient for identifying urban built-up areas of single data are 72.31% and 0.66, respectively, while the highest accuracy value and Kappa coefficient for urban built-up areas extracted from the fusion of night-time light data and POI data are 91.12% and 0.89, respectively, which proves that there is a limitation of insufficient accuracy in the extraction of single data in urban built-up areas. The fusion of night-time light data and POI data significantly improves the extraction accuracy of urban built-up areas.

By comparing the extraction accuracy of urban built-up areas in Kunming and Guangzhou, it can be found that before data fusion, the highest extraction accuracy and Kappa coefficient of urban built-up areas in Kunming from single-source data are 72.31% and 0.63, and the highest extraction accuracy and Kappa coefficient of urban built-up areas in Guangzhou are 72.20% and 0.66. Meanwhile after data fusion, the accuracy and Kappa coefficient of urban built-up area extraction in Kunming are 90.87% and 0.89, and the accuracy and Kappa coefficient of urban built-up area extraction in Guangzhou are 91.12% and 0.87. The comparison results of other urban cases show that after data fusion, the extraction accuracy of urban built-up areas increases to more than 90%, and the Kappa coefficient also increases to more than 0.85, indicating that data fusion has significant advantages in the extraction of urban built-up areas. Therefore, the method proposed in this study can also be used in a wider range of urban space research.

# 4. Discussion

In this study, POI data, Luojia1-01 data and the fused data are used to identify the urban built-up area of the main urban area of Kunming. This study holds the idea that the urban built-up areas identified by the POI data and Luojia1-01 data, respectively, based on the density-graph and GGM functions of the OSTU algorithm all supplement the deficiencies of the previously identified data and methods of urban built-up areas. First, it is reliable to identify urban built-up areas using POI big data based on the density-graph, which objectively eliminates the subjective thresholds set arbitrarily in the previous data and the scale effect under the spatial scale and the case study of Kunming proves the value of POI big data in exploring urban space, especially in urban built-up areas. Second, the night-time light data of Luojia1-01 is used to identify urban built-up areas, which on the one hand makes up for the lack of research on night-time light data in small and medium-sized cities due to the spatial resolution and, on the other hand, makes a case study on the use of Luojia1-01 data. The conclusion of this study can be further extended to a large number of cities in China and provide effective practical experience for the study of urban built-up areas.

In addition, a fusion data LJ&POI data is proposed in this study based on POI data and Luojia1-01 data, which provides a new research perspective and paradigm for the study of urban built-up areas. Both the ideas and methods of urban space research are greatly changed. Last but not least, the new fusion data has also achieved good results in the practical application of extraction studies of urban built-up areas, which has played a good role in the future research of urban built-up areas. However, compared with the traditional data, there is still little research on the fusion of data, which needs to be further deepened in theory and supplemented in practice. This is a process of continuous exploration.

On the one hand, compared with other urban built-up area extraction studies, this study proposes a new method based on image recognition (OSTU) to extract urban built-up area with night-time light data. In addition, this study also uses geometric mean to fuse night-time light data and POI data to extract more accurate urban built-up area, which supplements the theory and practical application of data fusion in urban built-up area extraction and thus plays a good role in promoting the study of urban built-up areas and urban space. On the other hand, compared with the latest studies on urban built-up areas [72], the accuracy of urban built-up areas extracted in this study is generally equivalent, which shows that the urban built-up areas extracted by the method proposed in this study have high accuracy and reliability [10,66]. What's more, compared with the study of urban built-up areas extracted from single-source data, this study still has a higher extraction accuracy [11].

In this study, the urban built-up area is extracted by the fusion of the Luojia1-01data and POI data. The Luojia1-01 data comes from the Luojia-1 experimental satellite of Wuhan University, China, and the POI data comes from Amap, both of which are open access. Extracting urban built-up areas by using open-access data is also the advantage of this research. As open data can be extended to other cities to extract urban built-up areas, this also indicates the repeatability and practicability of this study. There is no doubt that the accuracy of the research data will also affect the accuracy of the final urban built-up area extraction. For example, the spatial resolution of Luojia1-01 data is 130 M, while other night-time light data such as NPP is 500 M, which will have a great impact on the accuracy of urban built-up areas extracted before and after data fusion. Although this problem has not been discussed in this study, it will be the focus of the next studies.

There is no doubt that there are still some deficiencies in this study which need to be further improved. First, in terms of the use of POI big data, in this study, the weighted weight of POI data in space is assumed to be consistent, but as a geographic entity point, the entity category represented by each point of POI should be unequal. If the spatial weights of POI data are considered separately, it is not discussed here whether there will be deviations from the existing results in the identification of urban built-up areas. Second, the data fusion method proposed in this study is not suitable for the analysis of long-term span evolution, the reason being that the time series of POI data is very short, but both the POI data and LJ&POI data of this study involve POI data. Therefore, in the following research, multiple data fusion should be properly considered for long-term monitoring of changes in urban built-up areas.

# 5. Conclusion

In the study of urban built-up areas, the extraction of urban built-up area based on night-time light data is often inaccurate due to the spill over effect of lights. In this study, a method based on geometric mean

value is proposed to fuse night-time light data and POI data on the basis of fine identification of urban built-up areas. In addition, random verification point analysis is carried out on the extracted urban built-up areas. The results show that the accuracy of the extracted urban built-up areas has been significantly improved after data fusion, in which the research accuracy has been improved by nearly 20% and the Kappa coefficient has been improved by nearly 0.2. Therefore, the fusion of data is particularly important in the extraction of urban built-up areas and the application of urban space. The extracted real urban built-up areas also have important practical significance for the subsequent urban planning and construction.

Ethics. This study does not involve any ethical issues.

Data accessibility. All the data involved in this study can be obtained on the website, including Luojia1-01 data and POI data. For better access for readers, I upload the data to: http://dx.doi.org/10.17605/OSF.IO/C6QUK.

Authors' contributions. Data curation, methodology, writing the original draft and review & editing had done by X.H.; formal analysis, software, visualization and writing the review & editing had done by Z.Z.; software, visualization and writing the review & editing had done by Z.Y.

Competing interests. We declare we have no competing interests.

Funding. This study is not supported by any fund.

Acknowledgements. Thanks for the Amap that provided data support.

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
