## [Peer Review File · Royal Society Open Science]

Review History

RSOS-201268.R0 (Original submission)

Review form: Reviewer 1

Is the manuscript scientifically sound in its present form?

Yes

Are the interpretations and conclusions justified by the results?

Yes

Is the language acceptable?

Yes

Do you have any ethical concerns with this paper?

No

Have you any concerns about statistical analyses in this paper?

Yes

Recommendation?

Major revision is needed (please make suggestions in comments)

Comments to the Author(s)

This paper is interesting, and the authors conducted a comparison of different data scenarios in terms of identifying urban built-up areas, including nighttime light data, POI data, and a fusion of these two. The Kunming city was used as the case study. The paper has certain merits. However, there are some major concerns associated with the structure, novelty, and conclusions.

1. More up-to-date references can be added to section 2.2 in regards to urban data. For example: ANPR sensor data (Tang, J., Wan, L., Nochta, T., Schooling, J. and Yang, T., 2020. Exploring Resilient Observability in Traffic-Monitoring Sensor Networks: A Study of Spatial-Temporal Vehicle Patterns. *ISPRS International Journal of Geo-Information*, 9(4), p.247.), Vehicle trajectory data (Liu, J., Han, K., Chen, X.M. and Ong, G.P., 2019. Spatial-temporal inference of urban traffic emissions based on taxi trajectories and multi-source urban data. *Transportation Research Part C: Emerging Technologies*, 106, pp.145-165.) etc.
2. Section 2.3 – Line 56-60: duplicated content. Please check!
3. Highlighting objectives or main purposes is fine, but please also consider to summarise the key contributions of the paper.
4. Give a reference to “One Belt and One Road”.
5. Please further justify “The reason why the research area is not expanded to other districts and counties except the main urban area is mainly due to the significantly lower area of urban built-up areas of other districts and counties than that of the main urban area” why did the authors particularly underscore this point in this section?
6. Figure 2 – it seems that the in-text referencing for figure (a) and (b) in section 3.2.1 – line 55 and section 3.2.2 – Line 2 are mistaken. Should swap the position of (a) and (b) in figure 2.
7. The data sources should be properly cited, not just appending web links in the parentheses
8. Give the full name of all acronyms at their first appearance, such as OSTU, GGM, DN value, etc.
9. All method-related subsections need more details and descriptions. Think about the non-technical readers. More information and an in-detail and clear description of methods might be helpful. For example, what are the details for the implementations of those methods?
10. Section 4.1.1 – Line 37: Please write “FIG. 3” in a proper style.
11. The authors should use high-resolution figures.
12. Results are a bit too brief. The authors could add more in-depth descriptive discussion to the figures and results.
13. It would be better to use statistical tests to verify the different accuracy and other numerical results for different scenarios. The authors could think of questions like – are the differences in the accuracy statistically significant between pairs of “just POI”, “just LJ data”, and “combined them together” scenarios?
14. Comment No. 12 could be further tested through adding one or two more case studies of other comparable Chinese cities. Besides, drawing strong conclusions based on just one city case is relatively less convincing. Thus, it is recommended to add more test cases.
15. It would be better to compare your results and findings with one/more previous work(s) which use different data and methods.
16. Discussion is a bit shallow. More deeper thoughts would be useful. For example, what are the benefits for different stakeholders? How is the reliability of the proposed methods? Can this be transferred or have a wider application for other cities, or other areas of research? How would the data quality affect the results and findings? How is the sensitivity of the results (authors could even test the sensitivity in the results), etc. All in all, there are a lot of aspects, useful and insightful aspects, that the authors could expand on, which the authors did not do in the current version. Thus, it is recommended to consider all those points in the revision.
17. In the Conclusion section, the authors claimed that “the study holds the idea that the fusion of data is more important than the selection of methods in the identification of urban built-

up areas." Be careful with such strong assertion. The effectiveness of data fusion also heavily and inherently rely on the data quality and fusion methods. Furthermore, you didn't use a method of exhaustion here in your results, so it is better to avoid such assertive and affirmative statements. 18. Last but not the least, please keep a consistent format for all the entries in the reference list. Please be more rigorous.

Review form: Reviewer 2

Is the manuscript scientifically sound in its present form?

No

Are the interpretations and conclusions justified by the results?

No

Is the language acceptable?

No

Do you have any ethical concerns with this paper?

No

Have you any concerns about statistical analyses in this paper?

No

Recommendation?

Reject

Comments to the Author(s)

See attachement (Appendix A).

Decision letter (RSOS-201268.R0)

Dear Dr He

The Editors assigned to your paper RSOS-201268 "Study on Identification of Urban Built-up Area Based on Nighttime Lighting and POI Data---A Case Study of Kunming, China" have made a decision based on their reading of the paper and any comments received from reviewers.

Regrettably, in view of the reports received, the manuscript has been rejected in its current form. However, a new manuscript may be submitted which takes into consideration these comments.

We invite you to respond to the comments supplied below and prepare a resubmission of your manuscript. Below the referees' and Editors' comments (where applicable) we provide additional requirements. We provide guidance below to help you prepare your revision.

Please note that resubmitting your manuscript does not guarantee eventual acceptance, and we do not generally allow multiple rounds of revision and resubmission, so we urge you to make every effort to fully address all of the comments at this stage. If deemed necessary by the Editors, your manuscript will be sent back to one or more of the original reviewers for assessment. If the original reviewers are not available, we may invite new reviewers.

Please resubmit your revised manuscript and required files (see below) no later than 18-May-2021. Note: the ScholarOne system will 'lock' if resubmission is attempted on or after this deadline. If you do not think you will be able to meet this deadline, please contact the editorial office immediately.

Please note article processing charges apply to papers accepted for publication in Royal Society Open Science (<https://royalsocietypublishing.org/rsos/charges>). Charges will also apply to papers transferred to the journal from other Royal Society Publishing journals, as well as papers submitted as part of our collaboration with the Royal Society of Chemistry (<https://royalsocietypublishing.org/rsos/chemistry>). Fee waivers are available but must be requested when you submit your manuscript (<https://royalsocietypublishing.org/rsos/waivers>).

Thank you for submitting your manuscript to Royal Society Open Science and we look forward to receiving your resubmission. If you have any questions at all, please do not hesitate to get in touch.

on behalf of Professor Weisi Guo (Associate Editor) and R. Kerry Rowe (Subject Editor)
openscience@royalsociety.org

Associate Editor Comments to Author (Professor Weisi Guo):

Associate Editor: 1

Comments to the Author:

Dear authors,

I was able to obtain reviews from two expert reviewers. They found that the paper had limited methodological novelty and many issues. However, I also recognise that a lot of good work went into the paper and I do believe there is scope in the future should you wish to seriously improve upon the current version.

Editor

Reviewer comments to Author:

Reviewer: 1

Comments to the Author(s)

This paper is interesting, and the authors conducted a comparison of different data scenarios in terms of identifying urban built-up areas, including nighttime light data, POI data, and a fusion of these two. The Kunming city was used as the case study. The paper has certain merits. However, there are some major concerns associated with the structure, novelty, and conclusions.

1. More up-to-date references can be added to section 2.2 in regards to urban data. For example: ANPR sensor data (Tang, J., Wan, L., Nochta, T., Schooling, J. and Yang, T., 2020. Exploring Resilient Observability in Traffic-Monitoring Sensor Networks: A Study of Spatial-Temporal Vehicle Patterns. *ISPRS International Journal of Geo-Information*, 9(4), p.247.), Vehicle trajectory data (Liu, J., Han, K., Chen, X.M. and Ong, G.P., 2019. Spatial-temporal inference of urban traffic emissions based on taxi trajectories and multi-source urban data. *Transportation Research Part C: Emerging Technologies*, 106, pp.145-165.) etc.
2. Section 2.3 – Line 56-60: duplicated content. Please check!
3. Highlighting objectives or main purposes is fine, but please also consider to summarise the key contributions of the paper.
4. Give a reference to “One Belt and One Road”.
5. Please further justify “The reason why the research area is not expanded to other districts and counties except the main urban area is mainly due to the significantly lower area of urban built-up areas of other districts and counties than that of the main urban area” why did the authors particularly underscore this point in this section?
6. Figure 2 – it seems that the in-text referencing for figure (a) and (b) in section 3.2.1 – line 55 and section 3.2.2 – Line 2 are mistaken. Should swap the position of (a) and (b) in figure 2.
7. The data sources should be properly cited, not just appending web links in the parentheses
8. Give the full name of all acronyms at their first appearance, such as OSTU, GGM, DN value, etc.
9. All method-related subsections need more details and descriptions. Think about the non-technical readers. More information and an in-detail and clear description of methods might be helpful. For example, what are the details for the implementations of those methods?
10. Section 4.1.1 – Line 37: Please write “FIG. 3” in a proper style.
11. The authors should use high-resolution figures.
12. Results are a bit too brief. The authors could add more in-depth descriptive discussion to the figures and results.
13. It would be better to use statistical tests to verify the different accuracy and other numerical results for different scenarios. The authors could think of questions like – are the differences in the accuracy statistically significant between pairs of “just POI”, “just LJ data”, and “combined them together” scenarios?
14. Comment No. 12 could be further tested through adding one or two more case studies of other comparable Chinese cities. Besides, drawing strong conclusions based on just one city case is relatively less convincing. Thus, it is recommended to add more test cases.
15. It would be better to compare your results and findings with one/more previous work(s) which use different data and methods.
16. Discussion is a bit shallow. More deeper thoughts would be useful. For example, what are the benefits for different stakeholders? How is the reliability of the proposed methods? Can this be transferred or have a wider application for other cities, or other areas of research? How would the data quality affect the results and findings? How is the sensitivity of the results (authors could even test the sensitivity in the results), etc. All in all, there are a lot of aspects, useful and insightful aspects, that the authors could expand on, which the authors did not do in the current version. Thus, it is recommended to consider all those points in the revision.
17. In the Conclusion section, the authors claimed that “the study holds the idea that the fusion of data is more important than the selection of methods in the identification of urban built-up areas.” Be careful with such strong assertion. The effectiveness of data fusion also heavily and inherently rely on the data quality and fusion methods. Furthermore, you didn’t use a method of exhaustion here in your results, so it is better to avoid such assertive and affirmative statements.
18. Last but not the least, please keep a consistent format for all the entries in the reference list. Please be more rigorous.

Reviewer: 2
Comments to the Author(s)
See attachment.

===PREPARING YOUR MANUSCRIPT===

Your revised paper should include the changes requested by the referees and Editors of your manuscript. You should provide two versions of this manuscript and both versions must be provided in an editable format:
one version identifying all the changes that have been made (for instance, in coloured highlight, in bold text, or tracked changes);
a 'clean' version of the new manuscript that incorporates the changes made, but does not highlight them. This version will be used for typesetting if your manuscript is accepted.

===PREPARING YOUR REVISION IN SCHOLARONE===

<https://royalsociety.org/journals/authors/author-guidelines/#supplementary-material> to include a suitable title and informative caption. An example of appropriate titling and captioning may be found at https://figshare.com/articles/Table_S2_from_Is_there_a_trade-off_between_peak_performance_and_performance_breadth_across_temperatures_for_aerobic_sc_ope_in_teleost_fishes_/3843624.

Author's Response to Decision Letter for (RSOS-201268.R0)

See Appendices B & C.

RSOS-210838.R0

Review form: Reviewer 1

Is the manuscript scientifically sound in its present form?

Yes

Are the interpretations and conclusions justified by the results?

Yes

Is the language acceptable?

Yes

Do you have any ethical concerns with this paper?

No

Have you any concerns about statistical analyses in this paper?

No

Recommendation?

Accept as is

Comments to the Author(s)

Thank you for the revision. The authors did a thorough job on addressing the comments.

Review form: Reviewer 2

Is the manuscript scientifically sound in its present form?

No

Are the interpretations and conclusions justified by the results?

Yes

Is the language acceptable?

Yes

Do you have any ethical concerns with this paper?

No

Have you any concerns about statistical analyses in this paper?

No

Recommendation?

Accept with minor revision (please list in comments)

Comments to the Author(s)

Dear authors,

I believe the new submission has been improved a lot, but it will still need to be improved to reach the publication standard. See below for detailed comments:

1. P16 L10: please check the reference list accordingly.
2. Figure 1: this gives very limited information. You can actually combine figure 1 and 2 and then make the figure 1 as the inset map. In the figure 2, can you change the legend range from low-high to exact light intensity?
3. P18: so why you only have Kunming map but also treat Guangzhou as a study area?
4. Figure 3 is quite confused. Can you combine the 2 flowcharts together and make it clear?
5. Figure 8: this is very hard to see the difference between the 4 figures. Can you make it clear where is the difference?
6. Table 2: You should add the equation about how to calculate these measures (recall, precision, F1-score).

Decision letter (RSOS-210838.R0)

Dear Dr He

On behalf of the Editors, we are pleased to inform you that your Manuscript RSOS-210838 "Extraction of Urban Built-up Area Based on the Fusion of Night-time Light Data and POI Data" has been accepted for publication in Royal Society Open Science subject to minor revision in accordance with the referees' reports. Please find the referees' comments along with any feedback from the Editors below my signature.

Please submit your revised manuscript and required files (see below) no later than 7 days from today's (ie 07-Jul-2021) date. Note: the ScholarOne system will 'lock' if submission of the revision is attempted 7 or more days after the deadline. If you do not think you will be able to meet this deadline please contact the editorial office immediately.

Kind regards,
Royal Society Open Science Editorial Office
Royal Society Open Science

on behalf of Professor Weisi Guo (Associate Editor) and R. Kerry Rowe (Subject Editor)
openscience@royalsociety.org

Reviewer comments to Author:

Reviewer: 1

Comments to the Author(s)

Thank you for the revision. The authors did a thorough job on addressing the comments.

Reviewer: 2

Comments to the Author(s)

Dear authors,

I believe the new submission has been improved a lot, but it will still need to be improved to reach the publication standard. See below for detailed comments:

1. P16 L10: please check the reference list accordingly.
2. Figure 1: this gives very limited information. You can actually combine figure 1 and 2 and then make the figure 1 as the inset map. In the figure 2, can you change the legend range from low-high to exact light intensity?
3. P18: so why you only have Kunming map but also treat Guangzhou as a study area?
4. Figure 3 is quite confused. Can you combine the 2 flowcharts together and make it clear?
5. Figure 8: this is very hard to see the difference between the 4 figures. Can you make it clear where is the difference?
6. Table 2: You should add the equation about how to calculate these measures (recall, precision, F1-score).

===PREPARING YOUR MANUSCRIPT===

===PREPARING YOUR REVISION IN SCHOLARONE===

Author's Response to Decision Letter for (RSOS-210838.R0)

See Appendices D & E.

Decision letter (RSOS-210838.R1)

Dear Dr He,

I am pleased to inform you that your manuscript entitled "Extraction of Urban Built-up Area Based on the Fusion of Night-time Light Data and POI Data" is now accepted for publication in Royal Society Open Science.

on behalf of Professor Weisi Guo (Associate Editor) and R. Kerry Rowe (Subject Editor)
openscience@royalsociety.org

Appendix A

Article Review:

Study on Identification of Urban Built-up Area Based on Nighttime Lighting and POI Data---A Case Study of Kunming, China

Recommendation:

Reject

General comments:

This article attempts to use nighttime lights satellite images and point of interest data to identify urban built-up area in Kunming, Yunnan, China. I believe many research has done similar things in the literature and I don't see a clear new contribution from this article. I list a few articles below for authors' reference.

Besides the innovation perspective, the article is also very poorly written with very low-quality figures, which make it impossible to improve and reach to the journal standard. The logic of the introduction and literature review is not clear as well.

Lou, G., Chen, Q., He, K., Zhou, Y., & Shi, Z. (2019). Using Nighttime Light Data and POI Big Data to Detect the Urban Centers of Hangzhou. *Remote Sensing*, 11(15), 1821.

Jiang, Z., Zhai, W., Meng, X., & Long, Y. (2020). Identifying Shrinking Cities with NPP-VIIRS Nightlight Data in China. *Journal of Urban Planning and Development*, 146(4), 04020034.

Some specific comments:

Figures: all the figures need to meet journal requirement such as 300 dpi resolution.

It looks like the authors use summary instead of abstract, and please read the journal requirement carefully before any submission.

P3 L26: I don't think identification of urban built-up is a very difficult question now.

Introduction section is too short and did not really review existing research on urban built-up extraction.

The authors used a lot of acronym in the article but did not really explain it at the first instance.

In terms of data fusion of POI and night lights data, a lot of research have used it before, and I don't think it is a major research barrier anymore.

At the end of the introduction, I don't see a clear explanation in terms of the necessity of this research.

Figure 1: better to use district map rather than satellite images.

Figure 2: This figure needs improvement. The POI data give no information. Night lights data looks fine but what the value range mean?

Figure 3: very crude figure and I don't know why it is needed.

Equations: all the equations should be in a single line rather than embedded in the text.

When you use existing methods such as OSTU, you will need to cite the original reference. Also, it will be good to have a flowchart to explain your methodology.

Appendix B

Point-by-point response to reviewer 1

We would like to extend our sincere thanks to the reviewers who have carefully read our paper and put forward many valuable comments. These comments are believed to have a great effect on the improvement of our paper. In the past few months, we have carefully read every comment and comprehensively revised them according to the full text. No matter whether the final paper is accepted or not, we would like to thank you for your valuable advice. The following specific content is a reply to your comments one by one.

Point 1: More up-to-date references can be added to section 2.2 in regards to urban data. For example: ANPR sensor data (Tang, J., Wan, L., Nochta, T., Schooling, J. and Yang, T., 2020. Exploring Resilient Observability in Traffic-Monitoring Sensor Networks: A Study of Spatial–Temporal Vehicle Patterns. *ISPRS International Journal of Geo-Information*, 9(4), p.247.), Vehicle trajectory data (Liu, J., Han, K., Chen, X.M. and Ong, G.P., 2019. Spatial-temporal inference of urban traffic emissions based on taxi trajectories and multi-source urban data. *Transportation Research Part C: Emerging Technologies*, 106, pp.145-165.) etc.

Response 1: It has been realized that there are still many key references missing in the study. For this reason, we re-read the relevant research. While reorganizing the literature review, we have added many updated relevant references, which contributes to the increasement to the number of references in the full text from 57 to 72. Reference 15 is added. The detailed information of the reference can be found in the last reference part of the study.

Point 2: Section 2.3 – Line 56-60: duplicated content. Please check!

Response 2: We are very sorry that we did not notice this mistake. We have corrected this mistake and checked the full text in detail so that there is no such mistake in the full text.

Point 3: Highlighting objectives or main purposes is fine, but please also consider to summarise the key contributions of the paper.

Response 3: It has been realized that it is not enough to only talk about the goals and objectives of this research and not summarize the main contributions in the last part of the introduction of this paper, so we have reworked this part to further highlight the main contributions.

Compared with other studies, this study first analyzes the case of Kunming, and then uses Guangzhou as a case for further supplementary analysis to explore the research and application of data fusion in urban built-up area extraction, which complements the theoretical and practical application of data fusion in urban built-up area extraction, and plays a good role in promoting the research of urban built-up area and urban space, it also has a practical and positive guiding role for the future urban planning and construction. (Line 63-68)

Point 4: Give a reference to “One Belt and One Road”.

Response 4: Although One Belt and One Road is a well-known initiative in China, as an international paper, we really need to give the reference of this initiative, which will make international readers accept it more quickly. Therefore, we add the following references:

Solangi H R, Gilal F G, Tunio M Z. One belt one road initiative: the origin, current status, and challenges of China-Pakistan economic corridor[J]. International Journal of Technology, Policy and Management, 2018, 18(4): 313-335.

Point 5: Please further justify “The reason why the research area is not expanded to other districts and counties except the main urban area is mainly due to the significantly lower area of urban built-up areas of other districts and counties than that of the main urban area” why did the authors particularly underscore this point in this section?

Response 5: In the part of study area, we did not describe the selection of research area enough, which may cause some misunderstanding. Therefore, we made a detailed supplement to this part, and the supplementary content is as follows:

According to the statistical yearbook of Yunnan province in 2019, Kunming covers an area of 21,473 km², with an urban built-up area of 437km². Although the main urban area of Wuhua District, Guandu District, Xishan District and Chenggong District is only 2,542km², the urban built-up area

reaches 434.4km², which is also the reason why the five main urban areas of Kunming City are selected as the study area (Figure 1). (Line 77-80)

Point 6: Figure 2 – it seems that the in-text referencing for figure (a) and (b) in section 3.2.1 – line 55 and section 3.2.2 – Line 2 are mistaken. Should swap the position of (a) and (b) in figure 2.

Response 6: This is our mistake that we did not consider the sequence of the pictures when placing them. In order to avoid such a problem, we deleted the original picture and changed the original picture 2 into Figure 2 and Table 1, so that the information of the chart is more readable and the information presented is more complete.

Point 7: The data sources should be properly cited, not just appending web links in the parentheses

Response 7: It has been realized that the part of data reference is not standardized enough, and the data link should not be directly placed after it. Therefore, we deleted this part and uploaded all the original data to *10.17605/OSF.IO/C6QUK*, so that the use of this data can be directly accessed without reference.

Point 8: Give the full name of all acronyms at their first appearance, such as OSTU, GGM, DN value, etc.

Response 8: There are many abbreviations in the paper, some non-standard abbreviations will cause obstacles to the reading of the paper, so we have standardized and adjusted the abbreviations of the full text, including GGM (green level and gradient mapping), DN (Digital Number), etc., and other abbreviations can be seen in the specific content. As for OSTU, it refers to the whole process of this method, not an abbreviation, so we did not change the abbreviation of it.

Point 9: All method-related subsections need more details and descriptions. Think about the non-technical readers. More information and an in-detail and clear description of methods might be helpful. For example, what are the details for the implementations of those methods?

Response 9: Although three methods are mentioned in this paper, the introduction of the method is not very detailed, which will make it difficult for readers who are not professional in the field to understand. Therefore, we researched relevant references in this field again and supplemented some details of the method:

According to the binarization characteristic (gray characteristic) of the image itself, OSTU algorithm can divide the image into two parts: segmentation scene (foreground) and reference scene (background). Moreover, this algorithm can further judge the segmentation threshold of the automatically selected image by binarization of the whole image. When the threshold of image segmentation is selected, the difference between foreground and background is the biggest, which also is the segmentation standard of OSTU algorithm [67]. (Line 123-128)

The calculation step of OSTU algorithm is to first calculate the gradient value $f(x,y)$ of each pixel of the image, and then calculate the average gradient value T of the pixels with the same gray level in the image. Finally, the gray value \bar{t} of the maximum difference between the foreground and the background after setting the function of the GGM function. The obtained grayscale \bar{t} is the optimal image segmentation threshold. (Line 149-152)

The solution steps of Density-Graph are mainly divided into two steps, the first step is to determine the relationship between the density value d and the theoretical radius increment $\Delta s_d \wedge (1/2)$, and to obtain the derivative of the theoretical radius increment $\Delta s_d \wedge (1/2)$. For the theoretical radius increment $\Delta s_d \wedge (1/2)$ is derived, the equation d holds in theory. If the equation is equal to 0, then the density curve is uniformly diffused outwards. However, it is well known that there is basically no uniform expansion mode in urban built-up area expansion. Therefore, if the equation is greater than 0, the density curve is diffused outwards; if the equation is less than 0, the density curve is contracted inwards. The second step is to judge the critical value of density graph calculation. Actually, urban is a very complex system. As most cities are non-uniform outward expansion from multiple city centers

or clusters of cities, the Kernel Density Curve of point elements changes dramatically inside the urban space, which is also the reason why the performance of the Density-Graph curve inside the city should be fluctuating. However, from the macro scale of the whole urban space, the fluctuation of the Density-Graph curve should have a critical point with global significance. When $\lim_{da} \frac{d(\Delta S_d \wedge (1/2))}{da} > r$ appears, that is, when the fluctuation of Density-Graph presents an irreversible growth trend, it can be considered that r is a critical point with global significance, which can also be considered as the critical value of urban built-up area. (Line 163-176)

Point 10: Section 4.1.1 – Line 37: Please write “FIG. 3” in a proper style.

Response 10: It is true that in the paper submitted at the beginning, Figure and Fig were used confusedly, which caused the reviewer's misunderstanding. I am very sorry for this. Therefore, we have readjusted all the charts in the full text, and unified adjustment has been made to the format and style of reference. The specific content can be seen in the study.

Point 11: The authors should use high-resolution figures.

Response 11: We are very sorry that our low-resolution images do not reflect the detailed information of the paper. In order to solve this problem, we have done two things. First, we have remade all the pictures to ensure that the quality of the pictures is greater than 330dpi, and then we uploaded all the original pictures to the RSOS submission system.

Point 12: Results are a bit too brief. The authors could add more in-depth descriptive discussion to the figures and results.

Response 12: Indeed, our description of the experimental results was a little simple, which prevented us from presenting different data and experimental results expressed in the paper, especially in the part of result verification part. Therefore, we re-compared the experimental results, and the supplementary results are as follows:

Accuracy verification is an important step to test the reliability of this method for urban built-up area extraction. In this study, the urban built-up areas of Guangzhou are extracted by the fusion of Guangzhou night-time light data and POI data, besides, the confusion matrix and overall accuracy are used to verify the reliability and practicability of data fusion for the extraction of urban built-up areas. The extracted urban built-up areas of Guangzhou and the confusion matrix and accuracy verification are shown in Table 3. (Line 329-333)

It can be seen from Table 3 that the highest accuracy value and Kappa coefficient for identifying urban built-up areas of single data are 72.31% and 0.66 respectively, while the highest accuracy value and Kappa coefficient for urban built-up areas extracted from the fusion of night-time light data and POI data are 91.12% and 0.89 respectively, which proves that there is a limitation of insufficient accuracy in the extraction of single data in urban built-up areas. The fusion of night-time light data and POI data significantly improves the extraction accuracy of urban built-up areas. (Line 336-341)

By comparing the extraction accuracy of urban built-up areas in Kunming and Guangzhou, it can be found that before data fusion, the highest extraction accuracy and Kappa coefficient of urban built-up areas in Kunming from single source data are 72.31% and 0.63, and the highest extraction accuracy and Kappa coefficient of urban built-up areas in Guangzhou are 72.20% and 0.66. While after data fusion, the accuracy and Kappa coefficient of urban built-up area extraction in Kunming are 90.87% and 0.89, and the accuracy and Kappa coefficient of urban built-up area extraction in Guangzhou are 91.12% and 0.87. The comparison results of other urban cases show that after data fusion, the extraction accuracy of urban built-up areas increases to more than 90%, and the Kappa coefficient also increases to more than 0.85, indicating that data fusion has significant advantages in the extraction of urban built-up areas. Therefore, the method proposed in this study can also be used in a wider range of urban space research. (Line 342-351)

Point 13: It would be better to use statistical tests to verify the different accuracy and other numerical results for different scenarios. The authors could think of questions like – are the differences in the accuracy statistically significant between pairs of “just POI”, “just LJ data”, and “combined them together” scenarios?

Response 13: In the part of result verification, we realized that the description was a bit confusing, so we verified the results again and verified the research results in detail by using statistical test. The specific contents are shown in Table 3.

Table 3. Confusion Matrix and Kappa Coefficient (Line 335)

It can be seen from Table 3 that the highest accuracy value and Kappa coefficient for identifying urban built-up areas of single data are 72.31% and 0.66 respectively, while the highest accuracy value and Kappa coefficient for urban built-up areas extracted from the fusion of night-time light data and POI data are 91.12% and 0.89 respectively, which proves that there is a limitation of insufficient accuracy in the extraction of single data in urban built-up areas. The fusion of night-time light data and POI data significantly improves the extraction accuracy of urban built-up areas. (Line 336-341)

By comparing the extraction accuracy of urban built-up areas in Kunming and Guangzhou, it can be found that before data fusion, the highest extraction accuracy and Kappa coefficient of urban built-up areas in Kunming from single source data are 72.31% and 0.63, and the highest extraction accuracy and Kappa coefficient of urban built-up areas in Guangzhou are 72.20% and 0.66. While after data fusion, the accuracy and Kappa coefficient of urban built-up area extraction in Kunming are 90.87% and 0.89, and the accuracy and Kappa coefficient of urban built-up area extraction in Guangzhou are 91.12% and 0.87. The comparison results of other urban cases show that after data fusion, the extraction accuracy of urban built-up areas increases to more than 90%, and the Kappa coefficient also increases to more than 0.85, indicating that data fusion has significant advantages in the extraction of urban built-up areas. Therefore, the method proposed in this study can also be used in a wider range of urban space research. (Line 342-351)

Point 14: Comment No. 12 could be further tested through adding one or two more case studies of other comparable Chinese cities. Besides, drawing strong conclusions based on just one city case is relatively less convincing. Thus, it is recommended to add more test cases.

Response 14: It is true that even though the research results made in this study are of high accuracy, it does not prove that it is also applicable to other cities. Therefore, other cities need to be

added for supplementary explanation. Therefore, we choose Guangzhou to further explain the results of this research:

In addition, as one of the cities with the fastest urbanization process in China [66], the urban built-up area of Guangzhou has reached 1300.01 km² by 2019. Therefore, selecting Guangzhou as the case of this study will undoubtedly help to further prove the reliability of the method and conclusion of this study. (Line 80-83)

By comparing the extraction accuracy of urban built-up areas in Kunming and Guangzhou, it can be found that before data fusion, the highest extraction accuracy and Kappa coefficient of urban built-up areas in Kunming from single source data are 72.31% and 0.63, and the highest extraction accuracy and Kappa coefficient of urban built-up areas in Guangzhou are 72.20% and 0.66. While after data fusion, the accuracy and Kappa coefficient of urban built-up area extraction in Kunming are 90.87% and 0.89, and the accuracy and Kappa coefficient of urban built-up area extraction in Guangzhou are 91.12% and 0.87. The comparison results of other urban cases show that after data fusion, the extraction accuracy of urban built-up areas increases to more than 90%, and the Kappa coefficient also increases to more than 0.85, indicating that data fusion has significant advantages in the extraction of urban built-up areas. Therefore, the method proposed in this study can also be used in a wider range of urban space research. (Line 342-351)

Other details can be referred in the study.

Point 15: It would be better to compare your results and findings with one/more previous work(s) which use different data and methods.

Response 15: The results of this study need to be compared with others to find the value of this study. Therefore, we sorted out the current international papers on the extraction of urban built-up areas and compared the results of this study with these studies to highlight the value of this study. The specific contents are as follows:

On the one hand, compared with other urban built-up area extraction studies, this study proposes a new method based on image recognition (OSTU) to extract urban built-up area with night-time light data. In addition, this study also uses geometric mean to fuse night-time light data and POI data to

extract more accurate urban built-up area, which supplements the theory and practical application of data fusion in urban built-up area extraction and thus plays a good role in promoting the study of urban built-up area and urban space. On the other hand, compared with the latest studies on urban built-up areas [72], the accuracy of urban built-up areas extracted in this study is generally equivalent, which shows that the urban built-up areas extracted by the method proposed in this study have high accuracy and reliability [11],[66]. What's more, compared with the study of urban built-up areas extracted from single-source data, this study still has a higher extraction accuracy [10]. (Line 375-384)

Other details can be referred in the study.

Point 16: Discussion is a bit shallow. More deeper thoughts would be useful. For example, what are the benefits for different stakeholders? How is the reliability of the proposed methods? Can this be transferred or have a wider application for other cities, or other areas of research? How would the data quality affect the results and findings? How is the sensitivity of the results (authors could even test the sensitivity in the results), etc. All in all, there are a lot of aspects, useful and insightful aspects, that the authors could expand on, which the authors did not do in the current version. Thus, it is recommended to consider all those points in the revision.

Response 16: It has been realized that the discussion in this study is relatively simple. It is inappropriate that there is no comparison with other studies and no discussion of the advantages of this research. Therefore, we combined with No.15 to further deepen the discussion, including the comparison of other research results, the value and expansion of this research, etc., The specific content is as follows:

On the one hand, compared with other urban built-up area extraction studies, this study proposes a new method based on image recognition (OSTU) to extract urban built-up area with night-time light data. In addition, this study also uses geometric mean to fuse night-time light data and POI data to extract more accurate urban built-up area, which supplements the theory and practical application of data fusion in urban built-up area extraction and thus plays a good role in promoting the study of urban built-up area and urban space. On the other hand, compared with the latest studies on urban built-up areas [72], the accuracy of urban built-up areas extracted in this study is generally equivalent, which

shows that the urban built-up areas extracted by the method proposed in this study have high accuracy and reliability [11],[66]. What's more, compared with the study of urban built-up areas extracted from single-source data, this study still has a higher extraction accuracy [10]. (Line 375-384)

In this study, the urban built-up area is extracted by the fusion of the LuoJia1-01 data and POI data. The LuoJia1-01 data comes from the LuoJia-1 experimental satellite of Wuhan University, China, and the POI data comes from Amap, both of which are open-access. Extracting urban built-up areas by using open access data is also the advantage of this research. As open data can be extended to other cities to extract urban built-up areas, which also indicates the repeatability and practicability of this study. There is no doubt that the accuracy of the research data will also affect the accuracy of the final urban built-up area extraction. For example, the spatial resolution of LuoJia1-01 data is 130M, while other night-time light data such as NPP is 500M, which will have a great impact on the accuracy of urban built-up areas extracted before and after data fusion. Although this problem has not been discussed in this study, it will be the focus of the next studies. (Line 385-393)

Point 17: In the Conclusion section, the authors claimed that “the study holds the idea that the fusion of data is more important than the selection of methods in the identification of urban built-up areas.” Be careful with such strong assertion. The effectiveness of data fusion also heavily and inherently rely on the data quality and fusion methods. Furthermore, you didn't use a method of exhaustion here in your results, so it is better to avoid such assertive and affirmative statements.

Response 17: It has been realized that there are some problems with the expression of conclusions. The main reason is that the research we have done is limited to the research of the entire subject, and we cannot draw some conclusions outside of this research subjectively. Therefore, we reorganized this research and re-described the conclusion part based on the research results of this research:

In the study of urban built-up areas, the extraction of urban built-up area based on night-time light data is often inaccurate due to the spill-over effect of lights. While, in this study, a method based on geometric mean value is proposed to fuse night-time light data and POI data on the basis of fine identification of urban built-up areas. In addition, random verification point analysis is carried out on the extracted urban built-up areas. The results show that the accuracy of the extracted urban built-up

areas has been significantly improved after data fusion, in which the research accuracy has been improved by nearly 20% and the Kappa coefficient has been improved by nearly 0.2. Therefore, the fusion of data is particularly important in the extraction of urban built-up areas and the application of urban space. The extracted real urban built-up areas also have important practical significance for the subsequent urban planning and construction. (Line 407-415)

Point 18: Last but not the least, please keep a consistent format for all the entries in the reference list. Please be more rigorous.

Response 18: As for the references in this study, we found that some references were in different formats, which aroused our attention. Therefore, we rearranged the references, including the newly added ones, to ensure that all references were in uniform citation formats.

Appendix C

Point-by-point response to reviewer 2

We would like to extend our sincere thanks to the reviewers who have carefully read our paper and put forward many valuable comments. These comments are believed to have a great effect on the improvement of our paper. In the past few months, we have carefully read every comment and comprehensively revised them according to the full text. No matter whether the final paper is accepted or not, we would like to thank you for your valuable advice. The following specific content is a reply to your comments one by one.

Point 1: Lou, G., Chen, Q., He, K., Zhou, Y., & Shi, Z. (2019). Using Nighttime Light Data and POI Big Data to Detect the Urban Centers of Hangzhou. *Remote Sensing*, 11(15), 1821.

Jiang, Z., Zhai, W., Meng, X., & Long, Y. (2020). Identifying Shrinking Cities with NPP-VIIRS Nightlight Data in China. *Journal of Urban Planning and Development*, 146(4), 04020034.

Response 1: It has been realized that the number and content of references in this study are not enough. Including these two references, we add 15 relevant research references, which makes the summary and introduction parts of this study more complete. The newly added literatures can be seen in the last reference part.

Point 2: Figures: all the figures need to meet journal requirement such as 300 dpi resolution.

Response 2: We are very sorry that our low-resolution images do not reflect the detailed information of the paper. In order to solve this problem, we have done two things. First, we have remade all the pictures to ensure that the quality of the pictures is greater than 330dpi, and then we uploaded all the original pictures to the RSOS submission system.

Point 3: It looks like the authors use summary instead of abstract, and please read the journal requirement carefully before any submission.

Response 3: At the beginning, we did not understand the difference between Abstract and Summary in the writing of the paper. We wrote Abstract in the first version of the paper. In the process of revision, we compared the RSOS paper and rearranged the summary of this study as follows:

The accurate extraction of urban built-up areas is an important prerequisite for urban planning and construction. As a kind of data that can represent urban spatial form, night-time light data has been widely used in the extraction of urban built-up areas. As one of the geographic open-source big data, POI (Point of Interest) data has a high spatial coupling with night-time light data, so researchers begin to explore the fusion of the two data in order to achieve more accurate extraction of urban built-up areas. However, the current research methods and theoretical applications of the fusion of POI data and night-time light data are still insufficient compared with the dramatically changing urban built-up areas, which needed to be further supplemented and deepened. This study proposes a new method to fuse POI data and night-time light data. The results before and after data fusion are compared, and the accuracy of urban built-up area extracted by different data and methods is analyzed. The results show that the data fusion can avoid the shortage of single data and effectively improve the extraction accuracy of urban built-up area, which is greatly helpful to supplement the study of data fusion in urban built-up area, and also can provide decision-making guidance for urban planning and construction. (Line 1-12)

Point 4: P3 L26: I don't think identification of urban built-up is a very difficult question now.

Response 4: It's true that the identification of urban built-up area is not a very difficult problem. We made too subjective judgment when writing the paper. Therefore, when we reorganized the paper, we deleted all the sentences that are too subjective in the paper, which made the paper look more rigorous. The specific modification content can be viewed in the full text.

Point 5: Introduction section is too short and did not really review existing research on urban built-up extraction.

Response 5: It has been realized that there is no complete review of the research on the extraction of urban built-up areas in the introduction part, which cannot be ignored for the research on urban built-up areas. Therefore, we have re-sorted the research on urban built-up areas, and revised the introduction part as follows:

The urban built-up area refers to the non-agricultural construction land with complete functions of municipal public facilities within the urban area. The urban built-up area is the main carrier of urban population activities in urban space and a prominent manifestation of the high concentration of population and economy [8],[9]. To some extent, the number of urban built-up areas determines the level of regional development. Therefore, accurate extraction of urban built-up areas is not only helpful to understand the current situation of urbanization [10],[11], but also has a great correlation with the development of cities [12]. With the acceleration of global urbanization, urban built-up areas expand rapidly in a short period of time, accompanied by urban problems such as land use change and population imbalance caused by urban built-up area expansion [13],[14]. Therefore, accurate extraction of urban built-up areas is also of great importance to alleviate urban problems in the process of urbanization [15],[16]. (Line 1-10)

Data fusion has been widely used in the research of cities and related disciplines because it can not only integrate the advantages of multiple sources of data, but also better reflect the outstanding advantages of single data [59]. It has been shown in the studies that night-time light data and POI data have a strong coupling in urban space [60], so the fusion of the two data has a good effect in the application of urban space, including urban space identification, urban built-up area extraction, etc. [61]. By comparing the research of single source data [62], it can be found that although the fusion of night-time light data and POI data can significantly improve the accuracy of urban spatial identification and urban built-up area extraction, the research on data fusion is still insufficient compared with the drastic changes of urban built-up areas [62]. Therefore, in order to extract the urban built-up area more accurately and supplement the related research of data fusion, this study attempts to extract the urban built-up area of Kunming, China by fusing night-time light data and POI data, and uses Guangzhou, China as a case for verification and analysis to explore the extraction accuracy of built-up areas in this study. Finally, a reliable method and path for the accurate extraction of urban built-up areas by the fusion of night light and POI data are obtained. Compared with other studies, this study first analyzes the case of Kunming, and then uses Guangzhou as a case for further supplementary analysis to explore the research and application of data fusion in urban built-up area extraction, which complements the theoretical and practical application of data fusion in urban built-

up area extraction, and plays a good role in promoting the research of urban built-up area and urban space, It also has a practical and positive guiding role for the future urban planning and construction.
(Line 51-68)

Point 6: The authors used a lot of acronym in the article but did not really explain it at the first instance.

Response 6: There are many abbreviations in the paper, some non-standard abbreviations will cause obstacles to the reading of the paper, so we have standardized and adjusted the abbreviations of the full text, including GGM (green level and gradient mapping), DN (Digital Number), etc., and other abbreviations can be seen in the specific content. As for OSTU, it refers to the whole process of this method, not an abbreviation, so we did not change the abbreviation of it.

Point 7: In terms of data fusion of POI and night lights data, a lot of research have used it before, and I don't think it is a major research barrier anymore.

Response 7: It's true that there are many studies discussing the study of two types of data fusion in urban space, but different studies have different focuses. This study focuses on the advantages of data fusion in the extraction of urban built-up areas, this further complements the research of data fusion in urban area, which is of certain theoretical and practical value.

Point 8: At the end of the introduction, I don't see a clear explanation in terms of the necessity of this research.

Response 8: It is true that in the process of submitting the paper before, we did not clearly explain the necessity of research in the introduction part, so we revised the last part of it, including clearly explaining the purpose, goal and necessity of the research, and the specific modifications are as follows:

Therefore, in order to extract the urban built-up area more accurately and supplement the related research of data fusion, this study attempts to extract the urban built-up area of Kunming, China by fusing night-time light data and POI data, and uses Guangzhou, China as a case for verification and analysis to explore the extraction accuracy of built-up areas in this study. Finally, a reliable method

and path for the accurate extraction of urban built-up areas by the fusion of night light and POI data are obtained. Compared with other studies, this study first analyzes the case of Kunming, and then uses Guangzhou as a case for further supplementary analysis to explore the research and application of data fusion in urban built-up area extraction, which complements the theoretical and practical application of data fusion in urban built-up area extraction, and plays a good role in promoting the research of urban built-up area and urban space, It also has a practical and positive guiding role for the future urban planning and construction. (Line 58-68)

Point 9: Figure 1: better to use district map rather than satellite images.

Figure 2: This figure needs improvement. The POI data give no information. Night lights data looks fine but what the value range mean?

Figure 3: very crude figure and I don't know why it is needed.

Response 9: It has been realized that the resolution of the figure and the information contained in the figure are not complete. Therefore, based on the revision of No.2, we revised all the figures and the information contained in the figure. Due to the length of the refutation letter, the specific part can be seen in all the revised pictures.

Point 10: Equations: all the equations should be in a single line rather than embedded in the text.

Response 10: There are many formulas involved in this study, but the format of several formulas is wrong, so we rechecked all the formulas and re-typeset the content and format of these formulas. For specific modifications, you can check the parts of each formula.

$$\nabla f = \left[\frac{\partial f}{\partial x} \frac{\partial f}{\partial y} \right] \quad (1) \quad (\text{Line 140})$$

$$\nabla f = \text{mag}(\nabla f) = [G_x^2 + G_y^2]^{\frac{1}{2}} = \left[\left(\frac{\partial f}{\partial x} \right)^2 + \left(\frac{\partial f}{\partial y} \right)^2 \right]^{\frac{1}{2}} \quad (2) \quad (\text{Line 142})$$

$$R_i = \{(x, y) | f(x, y) = i\} \quad (3) \quad (\text{Line 146})$$

$$(i) = \frac{\sum_{(x,y) \in R_i} G(x,y)}{n_i} \quad i \in [0, 1, \dots, L-1] \quad (4) \quad (\text{Line 148})$$

$$\lim \frac{d(\Delta S_d \wedge (1/2))}{dd} > r \quad (5) \quad (\text{Line 178})$$

$$POIDN_i = \sqrt{POI_i \times DN_i} \quad (6) \quad (\text{Line 194})$$

Point 11: When you use existing methods such as OSTU, you will need to cite the original reference. Also, it will be good to have a flowchart to explain your methodology.

Response 11: It has been realized that when quoting documents, it did not take into account that the OSTU method is less involved in previous studies. Then the original document must be found for the quotation of this method, which will make the application of the method more credible, so we found the first referenced literature in this research as the original reference: **Pan, L., & Liu, Z. X. 2012. Automatic Airport Extraction Based on Improved Fuzzy Enhancement. In Applied Mechanics and Materials (Vol. 130, pp. 3421-3424). Trans Tech Publications Ltd.**

In addition, it has been realized that this study is relatively complex on the whole, including various data, methods and verification parts. Therefore, a clear flow chart is needed to clarify this study. Therefore, a new flow chart is added in Figure 4 to make the framework of this study clearer.

Appendix D

Point-by-point response to reviewer 1

Thank you very much for your approval of our revision

Appendix E

Point-by-point response to reviewer 2

Thank you very much for your approval of our revision. We have also made point-to-point revisions to the problems you mentioned in the second round.

Point 1: P16 L10: please check the reference list accordingly.

Response 1: I'm very sorry for this problem. We have corrected it. Please check page 16.

Point 2: Figure 1: this gives very limited information. You can actually combine figure 1 and 2 and then make the figure 1 as the inset map. In the figure 2, can you change the legend range from low-high to exact light intensity?

Response 2: Yes, we found that the content expressed in Figure 1 is too little. We modified Figure 1 and Figure 2 according to your opinions, and the revised figure is as follows:

Point 3: P18: so why you only have Kunming map but also treat Guangzhou as a study area?

Response 3: I'm very sorry that this is caused by our negligence. We have deleted and modified the expressions here.

Point 4: Figure 3 is quite confused. Can you combine the 2 flowcharts together and make it clear?

Response 4: We found that the flowchart was not very clear and did not express our meaning, so we redrew the flowchart. The new flowchart is as follows:

Point 5: Figure 8: this is very hard to see the difference between the 4 figures. Can you make it clear where is the difference?

Response 5: We have read your opinion carefully, and we think it is very important. After careful check, we find that there is no need for the existence of Figure 8, which will affect the understanding of this part, because the context clearly explains the differences between different data results, and there are specific tables and figures. However, it was not clear in the picture, so we deleted the picture 8 after discussion.

Point 6: Table 2: You should add the equation about how to calculate these measures (recall, precision, F1-score).

Response 6: Yes, we found that we did not mention the formula problem in this aspect in the full text, which was wrong, so we added the formula content of this part and modified it as follows:

The fraction is the harmonic average of recall and precision with a value range of 0~1. The higher the value is, the higher the precision is, the recall rate, precision rate and F_1 score are respectively:

$$precision = \frac{a_{overlap}}{a_{computed}} \quad (6)$$

$$recall = \frac{a_{overlap}}{a_{comparative}} \quad (7)$$

$$F_1 = \left(\frac{2}{\text{recall}^{-1} + \text{precision}^{-1}} \right) = 2 \frac{\text{precision} \cdot \text{recall}}{\text{precision} + \text{recall}} \quad (8)$$

a_{overlap} stands for the total area of the overlapped part of the built-up area and the reference built-up area, a_{computed} for the total area of built-up area extracted, $a_{\text{comparative}}$ is the total area of the reference built-up area. (Line 191-194)